# Microscopic theory of colour in lutetium hydride

Sun-Woo Kim [1]✉, Lewis J. Conway [1,2], Chris J. Pickard [1,2],
G. Lucian Pascut [3] & Bartomeu Monserrat [1,4]✉

Nitrogen-doped lutetium hydride has recently been proposed as a near-ambient-conditions superconductor. Interestingly, the sample transforms from blue to pink to red as a function of pressure, but only the pink phase is claimed to be superconducting. Subsequent experimental studies have failed to reproduce the superconductivity, but have observed pressure-driven colour changes including blue, pink, red, violet, and orange. However, discrepancies exist among these experiments regarding the sequence and pressure at which these colour changes occur. Given the claimed relationship between colour and superconductivity, understanding colour changes in nitrogen-doped lutetium hydride may hold the key to clarifying the possible superconductivity in this compound. Here, we present a full microscopic theory of colour in lutetium hydride, revealing that hydrogen-deficient $LuH_2$ is the only phase which exhibits colour changes under pressure consistent with experimental reports, with a sequence blue-violet-pink-red-orange. The concentration of hydrogen vacancies controls the precise sequence and pressure of colour changes, rationalising seemingly contradictory experiments. Nitrogen doping also modifies the colour of $LuH_2$ but it plays a secondary role compared to hydrogen vacancies. Therefore, we propose hydrogen-deficient $LuH_2$ as the key phase for exploring the superconductivity claim in the lutetium-hydrogen system. Finally, we find no phonon-mediated superconductivity near room temperature in the pink phase.

The proposal by Ashcroft that hydrogen-rich compounds could host high-temperature phonon-mediated superconductivity under pressure[1,2] has stimulated a profusion of theoretical proposals for high-pressure superconducting hydrides[3–9] and the subsequent experimental discovery of some of these[10–12]. This new class of hydride superconductors has re-ignited the search for superconductivity at ambient conditions, and Dasenbrock-Gammon and co-workers have recently reported superconductivity in nitrogen-doped lutetium hydride with a maximum critical temperature of 294 K at a moderate pressure of 10 kbar[13]. Interestingly, superconductivity is reported to

coincide with drastic colour changes in the reflectivity of the sample: increasing pressure transforms a non-superconducting blue metal to a superconducting pink metal at 3 kbar, and a further transformation to a non-superconducting red metal above 30 kbar.

This remarkable report has sparked a growing number of experimental[14–24] and theoretical[25–30] investigations, most of which have so far failed at reproducing or explaining near-ambient superconductivity. On the experimental front, most measurements of resistivity and magnetic susceptibility find no superconductivity near ambient conditions, with the exception of a recent work in which

[1]Department of Materials Science and Metallurgy, University of Cambridge, 27 Charles Babbage Road, Cambridge CB3 0FS, UK. [2]Advanced Institute for Materials Research, Tohoku University, 2-1-1 Katahira, Aoba, Sendai 980-8577, Japan. [3]MANSiD Research Center and Faculty of Forestry, Stefan Cel Mare University (USV), Suceava 720229, Romania. [4]Cavendish Laboratory, University of Cambridge, J. J. Thomson Avenue, Cambridge CB3 0HE, UK. ✉e-mail: swk38@cam.ac.uk; bm418@cam.ac.uk

resistivity changes consistent with high-temperature super-conductivity are reported[24]. Puzzlingly, multiple studies report pressure-driven colour changes, but these include a wide range of seemingly incompatible colour sequences and pressure conditions: blue-to-pink at 3 kbar and pink-to-red at 30 kbar in the original report[13]; blue-to-pink at 22 kbar and pink-to-red at 40 kbar[14]; blue-to-violet upon contact with a diamond culet, violet-to-red at 30 kbar and red-to-orange at 120 kbar[15]; blue-to-violet at 94 kbar[15]; blue-to-violet at 120 kbar, violet-to-pink-to-red gradually between 160 kbar and 350 kbar and red persisting up to 420 kbar[16,17]; blue-to-violet-to-pink-to-red with transition pressures differing by up to 60 kbar depending on the pressure medium used[18]; and persistent blue colour up to 65 kbar[19]. Growing evidence suggests that the colour changes are significantly affected by the initial compression procedure[15] and by the pressure medium used in the diamond anvil cell[18].

On the theoretical front there have been multiple reports of structure searches in the Lu–H binary and the Lu–H–N ternary systems[25–29]. Most studies only report metastable ternary structures, but Ferreira and co-workers report a ternary $Lu_4N_2H_5$ stable structure[29]. The roles of pressure and nitrogen doping[30] and of quantum and thermal ionic vibrations[31] in stabilising the cubic $LuH_3$ structure have also been studied. None of the predicted stable or metastable structures are found to be phonon-mediated superconductors near room temperature.

Given the claimed association between superconductivity and colour changes in the original superconductivity report, understanding colour changes in nitrogen-doped lutetium hydride holds the key to clarifying the possible superconductivity in this compound. However, experimental reports provide an inconsistent picture regarding pressure-driven colour changes, and there are no theoretical studies yet. In this work, we provide a full microscopic theory of colour in lutetium hydride.

## Results

### LuH₂ under ambient conditions

Lutetium hydride under ambient conditions crystallises in the $LuH_2$ stoichiometry with cubic space group $Fm\overline{3}m$. As shown in Fig. 1, $LuH_2$ adopts the fluorite structure with the lutetium atoms occupying the sites of an fcc lattice, and the hydrogen atoms occupying the tetrahedral interstitial sites.

$LuH_2$ is a metal whose reflectivity endows it with a blue appearance. We demonstrate the validity of our computational approach by reporting the calculated colour of $LuH_2$ at ambient conditions in Fig. 1. We show results using three distinct computational models to explore the potential role of electron correlation due to the presence of lutetium $5d$ electrons and the potential role of strong electron–phonon coupling due to the presence of hydrogen.

The first model we consider uses semilocal density functional theory (DFT) in the generalised-gradient approximation, labelled DFT in Fig. 1. This model provides a basic description of the electronic structure without a detailed treatment of electron correlation and without the inclusion of electron–phonon effects. The calculated reflectivity is large in the infrared region above 800 nm, is strongly suppressed in the red part of the visible spectrum with a calculated minimum at 710 nm, and increases gradually towards the blue part of the visible spectrum. The overall shape of the reflectivity is consistent with that observed experimentally[15] and directly leads to the blue colour of $LuH_2$.

Lutetium has an electronic configuration with a partially filled $5d$ shell, which suggests that electronic correlation beyond that captured by standard DFT may contribute to the electronic properties of $LuH_2$. To explore the possible role of electron correlation, we repeat our calculations using DFT corrected with a Hubbard $U$ term, labelled as DFT + $U$ in Fig. 1, which captures static correlation. The reflectivity curve has a similar shape to that obtained at the DFT level, but the

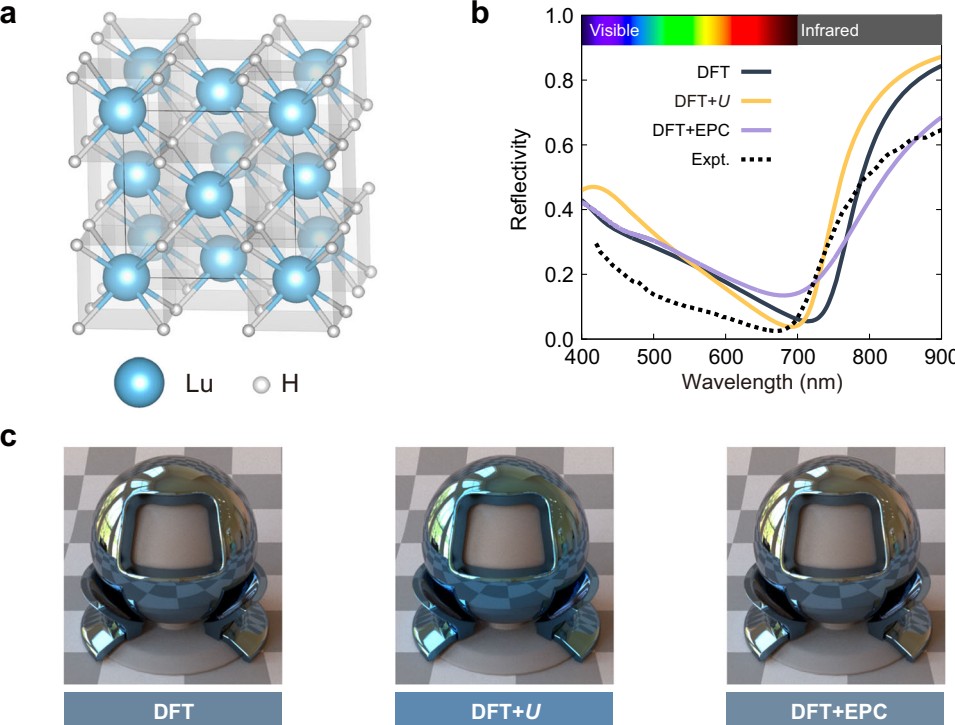

**Fig. 1 | Structure, reflectivity, and colour of LuH₂. a** Crystal structure of $Fm\overline{3}m$ LuH₂. **b** Reflectivity of LuH₂ calculated using semilocal density functional theory (DFT), DFT corrected with a Hubbard $U$ term (DFT + $U$), and DFT including electron–phonon coupling (DFT + EPC). The experimental reflectivity is taken from ref. 15. **c** Colour and photorealistic rendering of LuH₂ calculated using DFT, DFT + $U$, and DFT + EPC. The photorealistic rendering is shown as LuH₂ surrounding a grey ball with an opening in the centre.

minimum reflectivity has a lower value and occurs at a slightly shorter wavelength of 690 nm. Combined with a sligly larger reflectivity in the blue part of the visible spectrum, we obtain a slightly brighter blue colour for $LuH_2$ using the DFT + $U$ model. These results indicate that static electron correlation arising from lutetium only plays a minor role in $LuH_2$. We have also performed dynamical mean field theory calculations that capture dynamical correlation and also find that they can be neglected. We rationalise these results by noting that $5d$ orbitals have a large electron bandwidth spanning multiple eV (see Supplementary Fig. S1) which implies that the spatial extent of the orbitals is large and the corresponding local correlations weak.

Hydrogen is the lightest of all elements, and as such it exhibits significant nuclear motion, even at zero temperature, due to quantum zero-point effects. This significant nuclear motion can lead to strong electron–phonon coupling, and this is indeed the prime motivation behind the proposal that hydrogen-rich compounds could be high-temperature phonon-mediated superconductors. To explore the possible role of electron–phonon interactions in $LuH_2$, we repeat our calculations including contributions from both zero-point quantum nuclear motion at 0 K and thermal nuclear motion at finite temperature, labelled DFT + EPC in Fig. 1. The reflectivity curve has a similar shape to those obtained with DFT and DFT + $U$, but exhibits a lower value in the infrared region and a larger value in the red region. We again obtain a blue colour, indicating that electron–phonon coupling does not significantly modify the reflectivity of $LuH_2$.

Overall, we find that electronic correlation and electron–phonon interactions make a small contribution, and that the main features of the reflectivity curve and the resulting blue colour of $LuH_2$ are correctly captured by semilocal DFT. Therefore, our subsequent discussion neglects electron correlation and electron–phonon interactions, but further details about these contributions are included in the Supplementary Information.

## Lutetium hydride colour changes under pressure

To build a microscopic theory of colour in lutetium hydride, we first explore the pressure-driven colour changes in the lutetium–hydrogen system. We have performed extensive structure searches for stoichiometries ranging from $LuH_0$ to $LuH_3$ at multiple pressures. The results are summarised in the convex hull diagrams depicted in Fig. 2a. At 0 kbar, the only thermodynamically stable structures are $LuH_2$ ($Fm\bar{3}m$ space group) and $LuH_3$ ($P\bar{3}c1$ space group; not cubic). There are multiple metastable structures in the entire composition space from $LuH_0$ to $LuH_3$ that are within 60 meV/atom of the convex hull. Increasing pressure leads to multiple additional stable structures with stoichiometries intermediate between $LuH_2$ and $LuH_3$. We highlight that substoichiometric $LuH_{2-\delta}$ structures are close to the convex hull at both 0 and 400 kbar (within 11 and 42 meV/atom, respectively, up to $\delta = 0.25$). We have also checked their dynamical stability (see Supplementary Fig. S14), so we expect that they can be accessible experimentally.

We show the calculated colour as a function of pressure for the most stable structure at each composition between $LuH_0$ and $LuH_3$ in Fig. 2b. Consistently with experimental observations, we find that pure lutetium has a silvery white colour[20] (see also Supplementary Fig. S17) and, as described above in Fig. 1, $LuH_2$ has a blue colour. Across the entire composition space, the only compositions that exhibit a blue colour at ambient pressure occur for stoichiometries close to $LuH_2$. Similarly, the only stoichiometries that exhibit a violet colour at high pressure are those close to $LuH_2$. Specifically, we find this trend is present for substoichiometric $LuH_{2-\delta}$, but not present for suprastoichimoetric $LuH_{2+\delta}$. We also note that $LuH_3$ has a grey-green colour at ambient pressure that becomes grey with increasing pressure.

Figure 2c shows reflectivity curves for selected stoichiometries in the range $LuH_0$ to $LuH_3$ at multiple pressures. We note that only the reflectivity of $LuH_2$ has a minimum in the red part of the visible

spectrum leading to an overall blue colour, as already discussed in Fig. 1 above. The reflectivities of all other compositions show relatively flat curves across the visible spectrum, which lead to colours with various tones of grey.

Overall, the results depicted in Fig. 2 show that the only compounds in the lutetium–hydrogen binary space that are blue at ambient conditions and violet at high pressure have stoichiometries close to $LuH_2$ with a moderate amount of hydrogen vacancies. This conclusion still holds when structures in the ternary lutetium–hydrogen-nitrogen system are considered. To demonstrate this, we perform extensive crystal structure searches in the full Lu-H-N ternary space (see Supplementary Fig. S12), and note that our structure searches are the only ones of all those published that identify a stable ternary compound at ambient pressure[29]. We have calculated the colour of this stable compound and also the colour of multiple other ternary compounds that are not on the convex hull but whose simulated X-ray diffraction data is consistent with experimental reports. We find that none of these structures exhibit colours that are consistent with experiment (see Supplementary Fig. S13).

These observations allow us to conclude that the colour changes in the lutetium–hydrogen–nitrogen system are dominated by the $LuH_2$ stoichiometry. In particular, we discard the $LuH_3$ composition proposed by Dasenbrock-Gammon and co-workers to explain high-temperature superconductivity[13] as this structure has a grey-green colour at all pressures. $LuH_2$ has also been identified as the relevant stoichiometry by comparing the calculated equation of state[28] and X-ray diffraction patterns[25,26,28,29] to experiment, and we also note a recent experimental work that uses $LuH_2$-based samples and that has successfully reproduced the Raman spectrum and colour sequence with pressure reported in the original work[16].

## Hydrogen-deficient $LuH_{2-\delta}$

The calculated colour changes from blue-to-violet in $LuH_2$ are also observed in multiple experiments[15–18], but they occur at different pressures in different experiments, ranging from 0 to at least 190 kbar, but possibly higher as some experiments only observe a blue phase. Furthermore, some experimental observations reveal additional colour changes with increasing pressure, which include pink[13,14,18], red[13–15,18] and orange[15]. The pink colour is particularly important as it is associated with the superconducting phase in the original report[13]. Based on these observations, we next explore the pressure-driven colour changes of substoichiometric $LuH_2$ in more detail.

We show the pressure evolution of the reflectivity and colour of $LuH_2$ and hydrogen-deficient $LuH_{1.875}$ and $LuH_{1.750}$ in Fig. 3. At ambient conditions $LuH_2$ has the reflectivity described in Fig. 1 and repeated in Fig. 3 with a minimum in the red part of the spectrum that leads to an overall blue colour. With increasing pressure, the reflectivity minimum shifts towards shorter wavelengths, and the reflectivity from the red part of the spectrum increases, in agreement with experiment[15]. This leads to a gradual colour change from blue-to-violet with increasing pressure. $LuH_2$ undergoes a structural phase transition at 732 kbar to a phase of space group $P4/nmm$, which has a grey colour with an orange-red hue (reflectivity and colour shown in Supplementary Fig. S7).

At ambient conditions, $LuH_{1.875}$ exhibits a reflectivity with a shape similar to that of $LuH_2$ but with the minimum occurring at somewhat shorter wavelenghts of 600 nm. The resulting colour is still blue. Similar to $LuH_2$, increasing pressure leads to an overall shift of the reflectivity minimum to shorter wavelengths and to an increase in the reflectivity in the red part of the spectrum. As a result we observe a blue-to-violet colour change at a pressure of about 100 kbar, significantly lower than the corresponding colour change in pure $LuH_2$ and in the experimentally observed pressure range. Increasing pressure further leads to a gradual transition to pink (peaking at about 300 kbar), followed by red (peaking at about 500 kbar) and tending towards orange approaching 1000 kbar. Therefore, hydrogen-

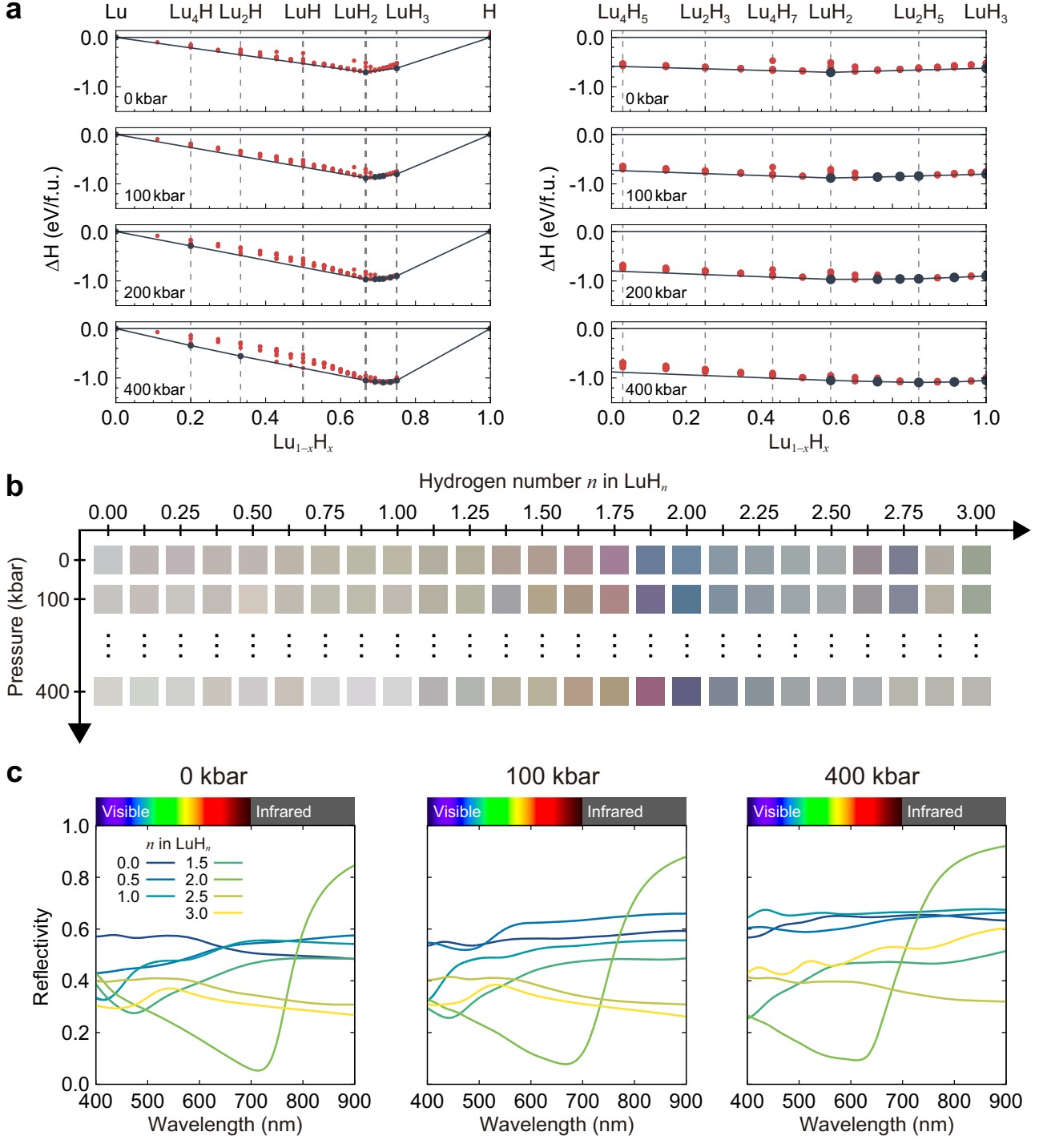

**Fig. 2 | Pressure dependence of the convex hull diagram, colour, and reflectivity of the lutetium–hydrogen binary system. a** Convex hull diagrams as a function of pressure for compositions in the range $LuH_0$ to $LuH_3$. Dark blue circles indicate thermodynamically stable structures and red circles indicate metastable structures. The dashed vertical lines for selected stoichiometries are shown for guidance only. **b** Colour of $LuH_n$ compounds as a function of pressure. **c** Reflectivity of $LuH_n$ compounds as a function of pressure.

deficient $LuH_2$ exhibits a sequence of colour changes that includes all colours reported experimentally.

The sequence and pressure of colour changes in hydrogen-deficient $LuH_2$ is strongly dependent on the concentration of hydrogen vacancies. Figure 3 also depicts the pressure evolution of the reflectivity and colour changes of $LuH_{1.750}$ with a higher concentration of hydrogen vacancies. In this case, the reflectivity minimum occurs at a wavelength of 550 nm at ambient conditions, giving a pink colour.

Increasing pressure suppresses the reflectivity in the blue region, turning the colour from pink towards orange at lower pressures than those necessary for $LuH_{1.875}$.

These results suggest that the seemingly contradictory experimental observations of colour changes in lutetium hydride are likely due to varying hydrogen vacancy concentrations in $LuH_2$. In particular, Dasenbrock-Gammon and co-workers observe a pink phase starting with a pressure of 3 kbar[13], significantly lower than the pressure

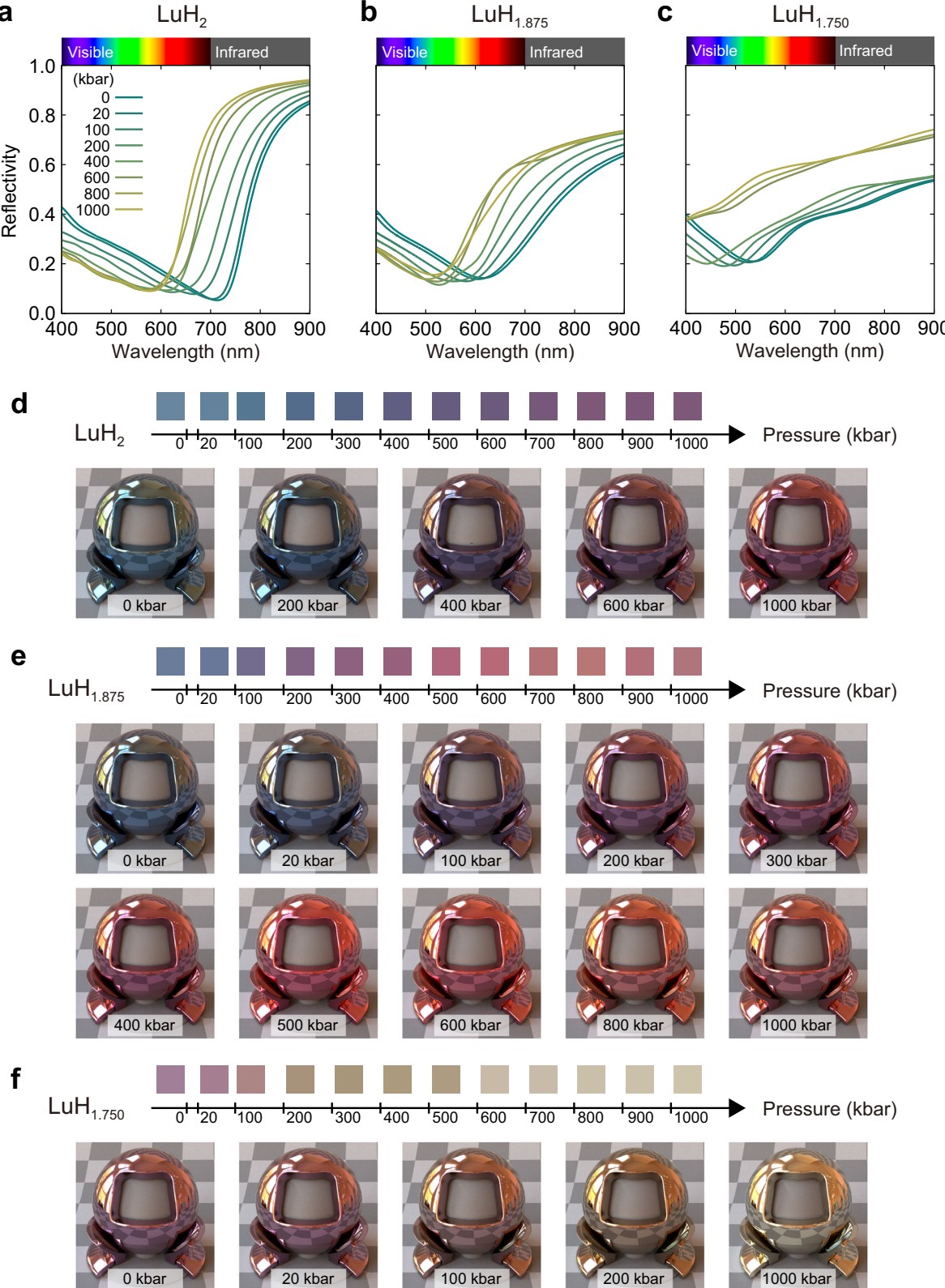

**Fig. 3 | Pressure dependence of the reflectivity and colour of pure and hydrogen-deficient lutetium dihydrides. a–c** Reflectivity as a function of pressure (in kbar) for **a** LuH$_2$, **b** LuH$_{1.875}$ and **c** LuH$_{1.750}$. **d–f** Colour and photorealistic rendering of **d** LuH$_2$, **e** LuH$_{1.875}$ and **f** LuH$_{1.750}$ as a function of pressure.

reported in multiple subsequent experiments. Our results suggest that this is due to a higher concentration of hydrogen vacancies in the original work compared to subsequent studies.

The concentration of nitrogen dopants also induces colour changes in LuH$_2$ (see Supplementary Fig. S10), but the colour changes driven

by nitrogen doping only play a secondary role compared to hydrogen vacancies. We have also tested the role that hydrogen vacancies and nitrogen doping have on the reflectivity and colour of cubic $Fm\bar{3}m$ LuH$_3$ (see Supplementary Fig. S11), as this phase has been tentatively identified as the parent phase responsible for the superconductivity claim[13].

**Table 1 | Calculated superconducting properties of hydrogen-deficient $LuH_2$ at various pressures**

| Structure | Pressure | $\lambda$ | $\omega_{\log}$ | $T_c^{AD}$ | $T_c^{E}$ |
|---|---|---|---|---|---|
| $LuH_{1.875}$ | 20 kbar | 0.333 | 257.516 | 0.107 | 0.190 |
| | 100 kbar | 0.326 | 257.997 | 0.085 | 0.161 |
| | 400 kbar | 0.360 | 243.566 | 0.220 | 0.328 |
| $LuH_{1.750}$ | 20 kbar | 0.277 | 231.482 | 0.008 | 0.035 |
| | 100 kbar | 0.276 | 224.351 | 0.007 | 0.032 |
| | 400 kbar | 0.299 | 211.951 | 0.023 | 0.062 |

$\lambda$ is the total electron–phonon coupling parameter computed from the Eliashberg function, $\omega_{\log}$ is the logarithmic averaged frequency, $T_c^{AD}$ is the superconducting critical temperature estimated from the semiempirical Allen–Dynes formula[54], and $T_c^{E}$ is the superconducting critical temperature estimated from the isotropic Eliashberg equation. A standard value of $\mu^* = 0.125$ is used for the Morel–Anderson Coulomb pseudopotential.

Our results show that the $LuH_3$ phase retains a grey colour under nitrogen doping, further discarding it as a relevant phase.

### Absence of phonon-mediated superconductivity in the lutetium–hydrogen–nitrogen system

Most experimental reports since the original announcement of near-ambient conditions superconductivity in the lutetium–hydrogen–nitrogen system have been unable to confirm this claim. Similarly, no calculation of stable and metastable phases in the lutetium–hydrogen-nitrogen system has predicted a high superconducting critical temperature within a phonon-mediated framework.

Our results suggest that hydrogen-deficient $LuH_2$ is responsible for the colour changes observed experimentally. Given the claim by Dasenbrock-Gammon and co-workers that it is the pink phase that is a room-temperature superconductor, we calculate the superconducting critical temperature of $LuH_{1.875}$ under pressure, which exhibits the pink phase. We find no room-temperature superconductivity, with a calculated critical temperature of the order of 0.1 K (Table 1; see also details in Supplementary Note 7).

## Discussion

Dasenbrock-Gammon and co-workers report superconductivity in nitrogen-doped lutetium hydride over the pressure range 3–30 kbar[13]. Importantly, the claimed pressure-driven transition to and from the superconducting phase occurs simultaneously with drastic colour changes in the sample, which is pink in the claimed superconducting phase, compared to blue below 3 kbar and red above 30 kbar. In addition, they attribute the claimed superconductivity to a cubic $LuH_3$ phase with some unknown concentration of nitrogen dopants and some unknown concentration of hydrogen vacancies.

Our calculations show that the cubic $LuH_3$ phase is not consistent with the colour changes observed experimentally. In addition, our results suggest that the only phase that is consistent with the observed colour changes is hydrogen-deficient cubic $LuH_2$, and that the concentration of hydrogen vacancies and nitrogen dopants controls the colour at each pressure. Finally, we also show that hydrogen-deficient $LuH_2$ is unlikely to be a high-temperature phonon-mediated superconductor.

Our work presents a compelling demonstration of how the first principles prediction of the colour of a material can be exploited to effectively identify the microscopic characteristics of the corresponding experimental samples. Given that colour is readily accessible experimentally, while other structural characterisation techniques can be challenging to implement (particularly under pressure), our work provides a promising new avenue for identifying composition and structure of complex samples. It would be interesting to further explore the applicability of this method to study the colour of other compounds, including strongly correlated materials[32,33] and magnetic materials[34,35].

## Methods

### Electronic structure calculations

We perform density functional theory (DFT) calculations using the Vienna *abinitio* simulation package (VASP)[36,37] implementing the projector-augmented wave (PAW) method[38]. We treat the $4f$ states of lutetium as valence by employing PAW pseudopotentials with 25 valence electrons ($4f^{14}5s^25p^65d^16s^2$). For the exchange-correlation energy, we use the generalised-gradient approximation functional of Perdew-Burke-Ernzerhof modified for solids (PBEsol)[39]. Converged results are obtained with a kinetic energy cutoff for the plane-wave basis of 400 eV and a **k**-point grid of size $40 \times 40 \times 40$ for the $LuH_2$ primitive cell and commensurate grids for other cell sizes and shapes (see convergence tests in Supplementary Fig. S20). The geometry of the structures is optimised until all forces are below 0.01 eV/Å and the pressure is below 1 kbar. We also perform select calculations using DFT corrected with a Hubbard $U$ term, and for these we use a value of $U = 3$ eV. We also perform select calculations using dynamical mean field theory with the EDMFT code[40,41], which implements density functional theory with embedded dynamical mean field theory (DFT + eDFMT). For the DFT part we have used the WIEN2K code[42].

### Reflectivity and colour

Our reflectivity calculations follow the methodology described in ref. 43. We calculate the complex dielectric function within the independent-particle approximation as implemented in VASP. In the optical limit ($\mathbf{q} \to 0$), the dielectric function $\varepsilon(\mathbf{q}, \omega)$ is given by the sum of an intraband Drude-like term $\varepsilon^{intra}(\mathbf{q}, \omega)$ due to the electrons at the Fermi surface and an interband term $\varepsilon^{inter}(\mathbf{q}, \omega)$ describing vertical transitions between valence and conduction bands. The explicit form of each term is given by[44,45]:

$$\varepsilon^{intra}(\mathbf{q}, \omega) = -\frac{\omega_D^2(\hat{\mathbf{q}})}{\omega(\omega + i\gamma)}, \tag{1}$$

where the independent-particle approximation Drude plasma frequency is

$$\omega_D^2(\hat{\mathbf{q}}) = \frac{4\pi}{V} \sum_{\mathbf{k}} \sum_n |\langle \psi_{n\mathbf{k}} | \hat{\mathbf{q}} \cdot \mathbf{v} | \psi_{n\mathbf{k}} \rangle|^2 \left( -\frac{\partial f_{n\mathbf{k}}}{\partial E_{n\mathbf{k}}} \right), \tag{2}$$

and

$$\varepsilon^{inter}(\mathbf{q}, \omega) = 1 - \frac{4\pi}{V} \sum_{\mathbf{k}} \sum_{n; n \neq n'} \sum_{n'} \frac{|\langle \psi_{n\mathbf{k}} | \hat{\mathbf{q}} \cdot \mathbf{v} | \psi_{n'\mathbf{k}} \rangle|^2}{(E_{n\mathbf{k}} - E_{n'\mathbf{k}})^2} \frac{f_{n\mathbf{k}} - f_{n'\mathbf{k}}}{\omega + E_{n\mathbf{k}} - E_{n'\mathbf{k}} + i\eta}. \tag{3}$$

In these equations, $V$ is the volume of the system, $|\psi_{n\mathbf{k}}\rangle$ is an electronic state with associated energy $E_{n\mathbf{k}}$ and labelled with quantum numbers $(n, \mathbf{k})$, $\mathbf{v}$ is the velocity operator, and $f_{n\mathbf{k}}$ is the Fermi–Dirac distribution. We use the empirical broadening parameters $\gamma = \eta = 0.1$ eV. To obtain the reflectivity, we average the dielectric function and the Drude plasma frequency as

$$\varepsilon(\omega) = \frac{\varepsilon(\hat{\mathbf{x}}, \omega) + \varepsilon(\hat{\mathbf{y}}, \omega) + \varepsilon(\hat{\mathbf{z}}, \omega)}{3} \text{ and } \omega_D^2 = \frac{\omega_D^2(\hat{\mathbf{x}}) + \omega_D^2(\hat{\mathbf{y}}) + \omega_D^2(\hat{\mathbf{z}})}{3}. \tag{4}$$

Using the relation $\varepsilon(\omega) = [n(\omega) + ik(\omega)]^2$ with the refractive index $n(\omega)$ and the optical extinction coefficient $k(\omega)$, we compute the reflectivity at normal incidence by assuming a vacuum-material interface as

$$R(\omega) = \frac{[n(\omega) - 1]^2 + k(\omega)^2}{[n(\omega) + 1]^2 + k(\omega)^2}. \tag{5}$$

We follow the method described in ref. 43 to obtain the colour from the reflectivity and we assign names to the calculated colours based on

the online tool ARTYCLICK (https://colors.artyclick.com/color-hue-finder). We note that the intensity of the colour can change slightly using different exchange-correlation functionals (see Supplementary Fig. S19).

We use the MITSUBA 3 renderer for the photorealistic rendering[46]. The photorealistic images presented in the main text are rendered by assuming an ideal bulk system with a clean surface. We have also considered the effects of surface roughness on the colour and find no significant changes (see Supplementary Fig. S18). We note that colour perception is subjective and that colour appearance depends on many effects including surface roughness and thickness of the sample. For this reason, we include all calculated dielectric functions as Supplementary Data, which allows readers to independently explore the resulting colours using their own setup.

### Electron–phonon coupling calculations

We use the finite displacement method in conjunction with non-diagonal supercells[47] to calculate the phonon frequencies $\omega_{\mathbf{q}\nu}$ and eigenvectors $\mathbf{e}_{\mathbf{q}\nu}$ of a phonon mode labelled by wave vector $\mathbf{q}$ and branch $\nu$. The electronic structure parameters are the same as those reported above, and we use a $4 \times 4 \times 4$ coarse $\mathbf{q}$-point grid to construct the matrix of force constants. Representative phonon dispersions are reported in Supplementary Note 6. We evaluate the imaginary part of the dielectric function at temperature $T$ renormalized by electron–phonon coupling using the Williams–Lax theory[48,49]:

$$\varepsilon_2(\omega; T) = \frac{1}{\mathcal{Z}} \sum_{\mathbf{s}} \langle \Phi_{\mathbf{s}}(\mathbf{u}) | \varepsilon_2(\omega; \mathbf{u}) | \Phi_{\mathbf{s}}(\mathbf{u}) \rangle e^{-E_{\mathbf{s}}/k_{\mathrm{B}}T}, \qquad (6)$$

where $\mathcal{Z}$ is the partition function, $|\Phi_{\mathbf{s}}(\mathbf{u})\rangle$ is a harmonic eigenstate $\mathbf{s}$ of energy $E_{\mathbf{s}}$, $\mathbf{u} = \{u_{\mathbf{q}\nu}\}$ is a vector containing all atomic positions expressed in terms of normal mode amplitudes $u_{\mathbf{q}\nu}$, and $k_{\mathrm{B}}$ is Boltzmann's constant. We evaluate Eq. (6) by Monte Carlo integration accelerated with thermal lines[50]: we generate atomic configurations in which the atoms are distributed according to the harmonic nuclear wave function in which every normal mode has an amplitude of $\left(\frac{1}{2\omega_{\mathbf{q}\nu}}[1 + 2n_{\mathrm{B}}(\omega_{\mathbf{q}\nu}, T)]\right)^{1/2}$, where $n_{\mathrm{B}}(\omega, T)$ is the Bose–Einstein factor. We note that the two terms in the Bose–Einstein factor imply that the electron–phonon renormalised dielectric function includes the effects of both quantum zero-point nuclear vibrations at $T = 0$ K and thermal nuclear vibrations at finite temperature. We build the electron–phonon renormalised reflectivity using the electron–phonon renormalised dielectric function.

For the superconductivity calculations, we evaluate the electronic and vibrational properties using QUANTUM ESPRESSO[51] together with GBRV ultrasoft pseudopotentials[52] and the PBE exchange-correlation functional[53]. We use a plane-wave cutoff of 50 Ry, a $4 \times 4 \times 4$ $\mathbf{k}$-point grid, and a Gaussian smearing of 0.02 Ry for geometry optimisations and electronic structure calculations. We use phonon $\mathbf{q}$-grids of size $4 \times 4 \times 4$ and dense electron $\mathbf{k}$-grids of size $8 \times 8 \times 8$ for the superconductivity calculations. In the calculation of the Eliashberg spectral function,

$$\alpha^2 F(\omega) = \frac{1}{2} \sum_{\nu} \int_{\mathrm{BZ}} \frac{d\mathbf{q}}{\Omega_{\mathrm{BZ}}} \lambda_{\mathbf{q}\nu} \delta(\omega - \omega_{\mathbf{q}\nu}), \qquad (7)$$

the integral is calculated by a sum over the $\mathbf{q}$-grids and the Dirac–delta functions are replaced by Gaussians with a width of 0.1 THz, where $\lambda_{\mathbf{q}\nu}$ is the electron–phonon coupling parameter for vibrational mode $\nu$ at $\mathbf{q}$. These parameters provide converged values of $\lambda$ and $\omega_{\log}$ derived from moments of $\alpha^2 F(\omega)$. We estimate the superconducting critical temperature $T_c^{\mathrm{AD}}$ using the McMillan–Allen–Dynes formula[54],

$$k_{\mathrm{B}} T_c^{\mathrm{AD}} = \frac{\omega_{\log}}{1.2} \exp\left[-\frac{1.04(1+\lambda)}{\lambda(1 - 0.62\mu^*) - \mu^*}\right], \qquad (8)$$

where $\mu^* = 0.125$. We also calculate $T_c^{\mathrm{E}}$ from a numeric solution to the isotropic Eliashberg equations. The values of $T_c$ do not change significantly for a range of $\mu^*$ values from 0.06 to 0.18.

### Structure searches

In the study of the lutetium–hydrogen binary system, we perform structure searches using the same Ephemeral Data-Derived Potential (EDDP)[55] as used by some of us in Ferreira and co-workers[29]. To generate the initial binary structures, we remove hydrogen atoms in cubic $LuH_3$. Starting with a $2 \times 2 \times 2$ supercell of the pristine cubic $LuH_3$ structure, we enumerate all possible symmetrically inequivalent defect structures with hydrogen vacancies using the DISORDER code[56,57]. This results in 55,066 unique structures for compositions in the range $LuH_0$ to $LuH_3$, reducing the original count from 17,777,214 through symmetry considerations. These structures are then optimised using the potential. The low energy structures resulting from the machine-learned potential relaxation are carried forward for subsequent DFT calculations.

We further use the ab initio random structure searching (AIRSS) method[58,59] to explore lutetium–hydrogen-nitrogen ternary compounds with cubic symmetry matching that reported in experimental X-ray diffraction data. We generate 1000 cubic structures by randomly replacing all Lu and H sites with H, N, or a vacancy, which supplements the 200,000 randomly generated Lu-H-N structures sampled in previous work[29]. We then relax these additional structures using DFT.

## Data availability

The data that support the findings of this study are available within the paper and Supplementary Information. In particular, the dataset for dielectric functions and structure files can be found at https://github.com/monserratlab/Lu_hydrides_colour.

## Code availability

The VASP code used in this study is a commercial electronic structure modelling software, available from https://www.vasp.at. The QUANTUM ESPRESSO code used in this research is open source: https://www.quantum-espresso.org/.

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

## Acknowledgements

S.-W.K. and B.M. are supported by a UKRI Future Leaders Fellowship [MR/V023926/1]. B.M. also acknowledges support from the Gianna Angelopoulos Programme for Science, Technology, and Innovation, and from the Winton Programme for the Physics of Sustainability. G.L.P. acknowledges funding from the Ministry of Research, Innovation, and Digitalisation within Program 1-Development of National Research and Development System, Subprogram 1.2-Institutional Performance-RDI Excellence Funding Projects, under contract no. 10PFE/2021. The computational resources were provided by the Cambridge Tier-2 system operated by the University of Cambridge Research Computing Service and funded by EPSRC [EP/P020259/1], by the UK National Super-computing Service ARCHER2, for which access was obtained via the UKCP consortium and funded by EPSRC [EP/X035891/1], and by the SCARF cluster of the STFC Scientific Computing Department.

## Author contributions

B.M. and S.-W.K. conceived the study and planned and supervised the research. S.-W.K. performed the DFT calculations, S.-W.K. and G.L.P. performed the DMFT calculations, L.J.C. and C.J.P. performed the structure searches, and all authors contributed to the analysis. B.M. and S.-W.K. wrote the manuscript with input from all authors.

## Competing interests

The authors declare no competing interests.
