## [Peer Review File · Nature Communications]

REVIEWER COMMENTS

Reviewer #1 (Remarks to the Author):

Very nice paper convincing explaining the peculiar color changes in Lu hydride and providing insight into its chemical composition and absence of superconductivity.

Reviewer #2 (Remarks to the Author):

The authors have addressed most of my questions and concerns. Still, I am not entirely satisfactory about their answer to my #3 comment. I believe what they said that the method [R6] is applicable to "a wide variety of metallic systems". However, I think metallic systems are simpler. It is more appreciable if they consider superconducting systems and magnetic materials, as exemplified by the 3 references I mentioned in my first round of review. I believe these examples, with correlations and spin interactions, are more closely related to the superconducting systems than the metallic systems in [R6].

The authors stated in their reply: "We also want to highlight the broader impact of our work for the wider scientific community. ...our work presents a compelling demonstration of how the first principles prediction of the colour of a material can be exploited to effectively identify the microscopic characteristics of the corresponding experimental samples." If they have calculations on the SC/magnetic materials I mentioned, their claim will be substantiated. Otherwise, it is not sure how broad their work has an impact--at least not including the strong correlation systems and many known high-Tc SC systems.

Authors say they have calculated 256,066 distinct structures; why don't they calculate 3 more? I am not sure how extensive the labor is. Will that cover 1/10 labor? Even if the labor is too high, the authors might want to point out some future work directions in their manuscript to justify the broader impact of their work. I dislike the current situation that the high-pressure calculation community try to keep away from strong correlations and magnetisms. If the authors can improve on this concern, I feel this work can be publishable in Nature Communications.

Summary of changes

Main text :

- We have added a new paragraph highlighting the significance of our research and discussing the potential applicability of our methodology to other compounds, including strongly correlated materials and magnetic materials

Added references

Please note that the following references have been added or updated to reflect recent publications. Citation numbers reflect the reference order in the most recent version of our manuscript:

- 26 Liu, M. et al. Phys. Rev. B 108, L020102 (2023).
- 28 Hilleke, K. P. et al. Phys. Rev. B 108, 014511 (2023).
- 29 Ferreira, P. P. et al. Nature Communications 14, 5367 (2023).
- 33 Tian, Y. C. et al. Phys. Rev. Lett. 116, 107001 (2016).
- 34 Wu, Q. et al. Chinese Physics Letters 37, 097802 (2020).
- 35 Zhao, J., Bragas, A. V., Lockwood, D. J. & Merlin, R. Phys. Rev. Lett. 93, 107203 (2004).
- 36 Zhao, J., Bragas, A. V., Merlin, R. & Lockwood, D. J. Phys. Rev. B 73, 184434 (2006).

Response to Reviewer 1

Reviewer (R) : Very nice paper convincing explaining the peculiar color changes in Lu hydride and providing insight into its chemical composition and absence of superconductivity.

Authors (A): We thank the Reviewer for their work reviewing our manuscript and for their positive feedback.

Response to Reviewer 2

Reviewer (R) : The authors have addressed most of my questions and concerns. Still, I am not entirely satisfactory about their answer to my 3 comment. I believe what they said that the method [R6] is applicable to “a wide variety of metallic systems”. However, I think metallic systems are simpler. It is more appreciable if they consider superconducting systems and magnetic materials, as exemplified by the 3 references I mentioned in my first round of review. I believe these examples, with correlations and spin interactions, are more closely related to the superconducting systems than the metallic systems in [R6]. The authors stated in their reply: “We also want to highlight the broader impact of our work for the wider scientific community. ...our work presents a compelling demonstration of how the first principles prediction of the colour of a material can be exploited to effectively identify the microscopic characteristics of the corresponding experimental samples.” If they have calculations on the SC/magnetic materials I mentioned, their claim will be substantiated. Otherwise, it is not sure how broad their work has an impact—at least not including the strong correlation systems and many known high-T_c SC systems. Authors say they have calculated 256,066 distinct structures; why don't they calculate 3 more? I am not sure how extensive the labor is. Will that cover 1/10 labor? Even if the labor is too high, the authors might want to point out some future work directions in their manuscript to justify the broader impact of their work. I dislike the current situation that the high-pressure calculation community try to keep away from strong correlations and magnetisms. If the authors can improve on this concern, I feel this work can be publishable in Nature Communications.

Authors (A): We are grateful to the Reviewer for their time and effort to revise our manuscript. And we are glad that the Reviewer is satisfied with our answer to most of their concerns in the previous round.

The one outstanding question refers to the applicability of our methodology to materials exhibiting strong correlation or magnetic order. We first note that we have performed DFT+U and DFT+DMFT calculations for the lutetium-hydrogen system to confirm that static and dynamical correlations are irrelevant in this system, so this questions does not affect the validity of our calculations here. Instead, the question more broadly addresses the wider applicability of our methodology to other compounds.

We agree with the Reviewer that it would be important to extend the present methodology to strongly correlated and magnetic compounds. Indeed, some of us are very much interested in modelling these phenomena, with examples of our recent work in this area including: Phys. Rev.

B 102, 245104 (2020) and Communications Physics 6, 45 (2023). Unfortunately, the computational and real time effort to explore this is not a simple fraction of the time we have dedicated to studying the lutetium-hydride system, it could very well require orders of magnitude more time. This is because strongly correlated phenomena and magnetism often require using methods beyond semilocal DFT, and they are invariably computationally more expensive and often require careful analysis of various physical parameters on a material-by-material basis. As such, we again emphasise that we agree with the referee and think such studies are very important, but they are well beyond the scope of the present work and we will pursue them independently.

To address the Reviewer's concern about the widest applicability of the methodology, we have modified the text to clarify that application to strongly correlated materials or magnetic materials will require further developments, and highlighted the materials and references that the Reviewer pointed to:

“... Our work presents a compelling demonstration of how the first principles prediction of the colour of a material can be exploited to effectively identify the microscopic characteristics of the corresponding experimental samples. Given that colour is readily accessible experimentally, while other structural characterisation techniques can be challenging to implement (particularly under pressure), our work provides a promising new avenue for identifying composition and structure of complex samples. It would be interesting to further explore the applicability of this method to study the colour of other compounds, including strongly correlated materials [33,34] and magnetic materials [35,36]...”

REVIEWERS' COMMENTS

Reviewer #2 (Remarks to the Author):

The authors addressed my last concern. It is nice to see the method has potential to extend to more complex material systems.